# The Emerging Role of Ablation in the Treatment of Primary and Metastatic Cancer of the Liver

**DOI:** 10.3390/jcm14228016

**Published:** 2025-11-12

**Authors:** Andrzej L. Komorowski

**Affiliations:** Department of Surgery, Faculty of Medicine, University of Rzeszów, al. Tadeusza Rejtana 16C, 35-959 Rzeszów, Poland; akomorowski@ur.edu.pl

**Keywords:** colorectal cancer, HCC, liver metastasis, liver resection, liver ablation, RFA, MWA

## Abstract

The traditional management of resectable colorectal liver metastases (CLMs) includes systemic therapy and curative (R0) surgery. Hepatocellular carcinoma (HCC) treatment options include R0 surgery, liver transplantation (LT), chemoembolisation, and targeted chemotherapy. Ablative techniques (radiofrequency ablation and microwave ablation) targeting liver lesions were until recently considered suitable only for patients deemed unfit for surgical resection. However, over time, data suggesting the non-inferior results of radical (A0) ablation compared with radical surgery have started to emerge. Given the lower complication rate of ablative therapies compared with surgery, the question arises as to whether ablation has the potential to replace the role of surgery in the treatment of HCC and colorectal cancer metastases to the liver. In this review, we address the current evidence on the topic and its possible impact on future clinical practice.

## 1. Introduction

The ablation of liver tumours with heat waves generated using radiofrequency (RFA) or microwave (MWA) can be performed either percutaneously, laparoscopically, or during an open procedure. The percutaneous approach can be performed as a day procedure under local anaesthesia. The laparoscopic approach is used frequently when the ablation zone is located close to the vital structures, while the open approach is commonly performed when ablation is accompanied by the resection of another liver lesion [1].

Surgical resection is currently considered the treatment of choice for colorectal cancer liver metastases (CLMs). It is characterised by 10-year overall survival rates reaching 24% when performed by experienced surgeons [2]. Virtually all CLM patients receive perioperative chemotherapy regimens at some point in their treatment, as it was previously shown that the use of systemic therapy has significantly increased and improved treatment outcomes [3]. Liver transplantation (LT), although showing very promising results, is still a rare treatment option for CLM patients due to significant organ shortages seen almost universally [4]. In recent decades, with the advent of ablative techniques, patients not suitable for surgery, patients operated on with salvage intent, and patients in whom an R2 (macroscopically non-radical) resection of CLM was performed have gained new treatment opportunities [5]. As the amount of evidence supporting the safe use of ablation rose, it also started to play a role in patients with small, centrally located CLM in order to spare as much of the healthy liver parenchyma as possible and thus optimise possibilities for future re-treatment. It was also introduced to treat patients requiring multiple resections in whom parenchyma sparing was of the utmost importance [6].

The encouraging results of the ablative treatment of CLM, with its superior surgical safety profile, led to implementing this technique in upfront resectable patients, allowing for the direct comparison of the results of ablation with resection in both retrospective and prospective randomised studies. If the non-inferior results of ablation were to be confirmed, the superior safety profile could make ablation a preferred treatment option for patients with CLM.

Hepatocellular carcinoma of the liver (HCC) is a leading cause of cancer-specific mortality, especially in Asia [7], and the sixth most common cancer worldwide [8]. Liver transplantation is the best treatment option, although due to organ shortage, it is available only for a fraction of HCC patients. The Barcelona Clinic Liver Cancer (BCLC) system is a well-established cornerstone for staging and treatment decision-making in patients with HCC, depending on the stage of local disease and underlying liver disease [9]. Patients fulfilling either the Milan, San Francisco, or other nationally accepted extended criteria are nevertheless listed for transplantation [10]. In order to keep patients on the waiting list without progression of the disease that would exclude the possibility of LT, several bridge therapies have been proposed. These include transarterial chemoembolisation (TACE), transarterial radioembolisation (TARE), radiofrequency ablation (RFA), and microwave ablation (MWA) [7]. In a comprehensive systematic review of available data published in 2013, it was found that the effects of RFA versus no intervention, chemotherapeutic treatment, or liver transplantation are unknown due to lack of data. There was only moderate-quality evidence suggesting that hepatic resection is superior to RFA as far as survival is concerned [11]. In the current version of the BCLC algorithm, ablation of HCC is indicated for very early and early HCC patients who do not qualify for surgery [9]. There is, however, a growing amount of evidence supporting the utility of the ablation technique for the treatment of more advanced HCC patients and recurrent tumours.

Since the safety profile of both RFA and MWA ablation techniques has been consistently confirmed to be superior to surgical resection, the role of ablation in the treatment of the two most common malignant liver tumours i.e., CLM and HCC, can be expected to grow. This review will attempt to show that ablative techniques can be successfully used not only in patients disqualified from surgical resection but also as a technique equivalent to or, in some cases, even better than surgery.

## 2. Current Evidence for Liver Ablation

### 2.1. Colorectal Cancer Liver Metastasis (CLM)

A meta-analysis of studies published before February 2015 comparing RFA with resection showed a significantly lower rate of complications after ablation, but also a lower survival rate and a higher rate of recurrence for patients treated with ablation [12]. Similar results were seen in a systematic review of 18 studies comparing RFA with resection published in 2018 [13]. The authors’ conclusion was that resection is still the gold standard in the treatment of CLM. A meta-analysis of 22 studies published before October 2020 showed that there was no significant difference between RFA and resection in 30-day mortality, with a pooled OR of 0.88 (95% CI 0.34–2.29; *p* = 0.80). CLM patients undergoing RFA experienced significantly higher incidences of marginal and intrahepatic recurrence than surgical patients, with pooled ORs of 7.09 (95% CI 4.56–11.2) and 2.02 (95% CI 1.24–3.28). In addition, RFA showed a lower 1-, 3- and 5-yr OS rate than resection with pooled ORs of 0.39, 0.40, and 0.60, respectively. A lower 5-year DFS rate was also found in RFA compared with the resected group, with a pooled OR of 0.74 (95% CI 0.56–0.97; *p* = 0.03; 1231) [14].

Looking at this data, it is important to note, however, that as resection was considered the treatment of choice for resectable CLM, the population of patients in retrospective studies receiving ablative treatment was by definition characterised by poorer long-term outcomes, as ablation was typically offered to patients deemed unsuitable for surgery. Therefore, since the results of the use of ablative therapy for unresectable CLMs are encouraging [15], it can be speculated that when offered to patients with better prognostic factors (upfront surgical candidates), it could yield results similar to surgery. The second important source of residual confounding in these studies is the use of RFA as a sole ablative therapy. The biological effect of MWA is believed to be superior to that of RFA. MWA requires less time and is more effective in the ablation of larger tumours and more radical in the peripheral parts of spherical tumours [16]. Furthermore, MWA is particularly advantageous for tumours located near large vessels, where the heat-sink effect can reduce the efficacy of other thermal ablation techniques, such as RFA [1]. Hence, the use of the term “ablation” in this context can be misleading as we are basing the indications for the use of MWA ablation on the results of RFA ablation. And, indeed, several studies have shown even more encouraging results on the use of MWA compared to those of RFA in the context of CLM.

As stated previously, the traditional indication for ablation of CLM was an unresectable lesion. Ablation has been used in combination with systemic chemotherapy to downsize tumours and convert previously unresectable CLM to resectable states. The combination of ablation, chemotherapy, and surgery has been associated with improved long-term survival. In patients who successfully underwent complete resection or ablation, the median survival reached 59 months, compared to 16 months in the case of those who did not [17].

In a significant series of over 3000 liver tumours, both resectable and nonresectable, treated with MWA, the median overall survival for CLM patients was 3.9 years, which is close to the results of surgical resection and at the same time is characterised by a better safety profile [18].

These results allowed for further direct comparisons of ablation with resection of CLM. An analysis of 184 patients with CLM treated between 2012 and 2017 showed that after propensity score matching, the 1-, 2-, and 3-year local tumour progression-free survival rates were found to be 60.3%, 19.1%, and 17.6% in the MWA group and 72.1%, 35.3%, and 26.5% in the resection group, respectively (*p* = 0.049). The 1-, 3-, 5-, and 7-year overall survival rates in the two groups were similar (*p* = 0.943). In the MWA group and resection group, the median hospital stay was 1 (0–12) days and 7 (1–27) days (*p* = 0.005), respectively; major complications occurred in 2% of MWA patients and 12.9% of surgical patients, which once again underlies the main advantage of ablation over resection, i.e., its safety profile [19].

In a similar study, which used data from the national registry in Sweden, it was shown that 3-year overall survival (OS) favoured resection over MWA (76 and 69%, *p* = 0.005). However, after propensity score matching (70 MWA patients, 201 resection patients), no difference in 3-year OS was shown between resected and ablated patients (76% and 76%, *p* = 0.253), with a median OS of 54.7 (95% confidence interval 48.6–60.9) months and 48 (40.1–56.1) months, respectively [20]. Furthermore, in a study reporting RFA performed between 2000 and 2018, in which propensity score match analysis was used, it was shown that after matching the groups, both groups showed a similar complication rate (17 vs. 33%; *p* = 0.18) and recurrence rate (58 vs. 64%; *p* = 0.09), without significant differences in 5-year DFS and OS (RFA compared with surgery, respectively, 25 vs. 37%; *p* = 0.09 and 42 vs. 53%; *p* = 0.09). There was, however, a difference in the size of CLM, with a mean tumour size of, respectively, 2.5 cm (0.8–6.5) in MVA and 3.4 cm (1–7.5) in the resected group [21].

Similar encouraging results were presented by the non-randomised registry-controlled MAVERRIC trial, which showed no differences in overall survival, while maintaining lower morbidity with thermal ablation compared with surgical resection [22].

In virtually all studies reporting the results of various ablative techniques for CLM compared with resection, there are consistently fewer complications and a faster recovery time compared with reported surgery. The statistically significant difference favouring RFA over resection complication-wise was estimated to have a relative risk ranging from 0.47 [23] to 0.44 [12]. Complications were found in 25% of surgical patients and in 10–12% of ablation patients. Shorter hospital stays were reported in a consistent manner by the majority of authors [12,13,23].

If the safety profile of ablative therapies is superior to surgery, then the rationale to offer surgery to patients lies in better oncological results. However, the data cited previously failed to confirm this decisively. While earlier studies, mainly comparing RFA with surgery, indeed showed the oncological superiority of surgery [12,13,14], some more recent data, which takes both MWA and RFA into account, has failed to do so [19,20,22]. In particular, the results of the multicentre MAVERRICK trial, which compared 98 patients with up to five resectable liver metastases smaller than 3 cm treated with microwave ablation (MWA) with matched patients who underwent surgical resection, showed that OS rates at 3 years were 78% after MWA versus 76% after resection (*p* = 0.861). Estimated 5-year OS rates were 56% (CI 45–66%) versus 58% (CI 50–66%). The adjusted hazard ratio for treatment type was 1.020 (CI 0.689–1.510). Overall and major complications were lower after MWA (percentage decrease 67% and 80%, *p* < 0.01). However, hepatic re-treatments were more frequent after MWA (percentage increase 78%, *p* < 0.01) [22].

The data given above cleared the way for a randomised trial that would compare ablation with resection of CLM. Such a multicentre study (COLLISION trial) was designed as a non-inferiority trial comparing the results of ablation with resection of small (up to 3 cm) CLMs. This trial was prematurely terminated after only 50% of the planned sample size enrolled, due to futility for efficacy endpoints and early evidence of a benefit in terms of safety, with superior safety outcomes in the ablation group and no difference in local control between the groups. Out of 145 patients in the ablation group, only 1 received RFA while 134 received MWA treatment. And while local control and median overall survival did not differ between the groups, patients in the experimental group had fewer adverse events than those in the control group (28 [19%] vs. 67 [46%]; *p* < 0·0001). In addition, serious adverse events occurred in 7% of patients in the experimental group and 20% in the resected group again, confirming the main argument for ablation [24].

On the basis of the data cited above, it is safe to say that the role of ablation of CLMs has been firmly established in cases of unresectable tumours, centrally located tumours that would otherwise require major hepatectomy, and as an adjunct to surgical resection for multiple tumours when it is of the utmost importance to spare liver parenchyma. There are initial data suggesting that ablation is non-inferior to surgery when facing a small (<3 cm) resectable CLM. In all clinical scenarios, it is characterised by a very favourable safety profile. As data, especially long-term oncological outcomes from various settings of CLM ablation using both MWA and RFA, continue to emerge, the indications for its use may expand substantially.

### 2.2. Hepatocellular Carcinoma (HCC)

LT remains the best available treatment for HCC, with RFS exceeding 90% for patients fulfilling the Milan criteria [25]. However, due to organ shortage and resulting long waiting lists for LT, other treatment options are considered, either as a bridge therapy for transplantation or as a curative approach. A well established and universally accepted staging and treatment algorithm for HCC is the Barcelona Clinic Liver Cancer (BCLC) system [9]. Based on this algorithm, depending on the Child–Pugh score, AFP level, size and number of intrahepatic lesions, and presence of portal invasion or extrahepatic spread, the patient falls into one of five categories: very early (0), early (A), intermediate (B), advanced (C), and terminal (D) stage. Treatment options include ablation, resection, LT, TACE, systemic treatment, and the best supportive care.

Almost 30% of patients undergoing bridge therapies before LT experience a pathological complete response. In this group of patients, the long-term oncological results are excellent, with hardly any local recurrences [7]. In one of the largest reported ablation series, median survival after MWA of HCC was 3.7 years [18]. The use of percutaneous RFA in a group of early and intermediate HCC showed even better results, with a median survival rate of 62 months and a 5-year survival rate of 52% [26]. These findings, together with relatively low morbidity after ablation of liver tumours [21,22], raise the question of the role of ablation (be it RFA or MWA) as the sole treatment of HCC.

It is important to note that the main indication for local ablation of HCC was, for a relatively long period of time, unresectability of the tumour, which suggests that the results of ablation apply only to those patients with long-term results inferior to those of surgery. Comparisons of different ablation techniques, namely RFA and MWA, among unresectable HCC patients showed that local recurrences after both techniques were similar but MWA was better than RFA in a subgroup of larger tumours. While the complete response rate was higher for MWA and 3-year survival better for RFA, these differences did not reach statistical significance [27].

Ablation of a small (<3 cm) HCC as a bridge therapy before LT is a widely used approach, hence the significant amount of data concerning this clinical scenario [9]. In a recent study comparing MWA with RFA for early-stage HCC, a large group of more than 1000 patients from a single centre were analysed. The results showed that the 1-, 2-, and 3-year recurrence rates were 40.89%, 68.07%, and 84.13% for MWA and 39.13%, 62.64%, and 75.71% for RFA. The overall survival rates at 1, 2, and 3 years were 70.47%, 40.98%, and 21.90% for MWA and 70.53%, 44.85%, and 30.60% for RFA. While this large-scale study was retrospective and the majority of patients underwent an RFA procedure (762 RFA patients vs. 266 MWA patients), its results prove that when performed by experienced hands. both RFA and MWA can yield very promising results [28].

A systematic review and meta-analysis of papers dealing with MWA vs. RFA treatment of larger HCC up to 5 cm reported that the efficacy of MWA, as measured by incomplete ablation and complication rates, was similar to that of RFA for HCC, i.e., less than 5 cm [29]. This could be explained by some centres’ greater experience with RFA compared to MWA, as the latter was introduced into clinical practice more recently [8]. As stated previously [16], the biological results of MWA in destroying liver tumours should theoretically be superior to those of RFA. In an experimental study, it was also shown in the context of HCC treatment that high-power MWA and RFA activate systemic immunity and enhance Th1 response in a murine model of HCC. If confirmed, this finding could further improve the results of ablation [30].

Based on the analysis of seven prospective trials analysing the effect of treatment of 921 patients, no difference between MWA and RFA was found in terms of complete response. Survival rates were also similar. The local recurrence rate was similar as well, but the distant recurrence rate was significantly lower with MWA (RR 0.60, 0.39–0.92). Disease-free survival at 1, 2, and 3 years did not differ between the groups, but at 5 years, MWA showed better results. The complication rate of both methods was similar, with bleeding and hematoma representing the most frequent complications [31].

When it comes to comparisons of ablation of HCC with a traditional surgical approach, in a retrospective study, a group of 424 patients with a solitary HCC of 3–5 cm were treated by resection or percutaneous RFA. The 5-year OS was comparable between the two modalities (*p* = 0.367), although 5-year DFS was significantly lower in the RFA group than in the LR group (*p* = 0.001). Interestingly, the two modalities did not differ in severe post-treatment complications, which is in stark contrast to other studies comparing ablation with resection [8]. This, in turn, can be explained by the significant experience of the reporting centre in liver resection.

In another analysis of 319 patients aged 65 or over with a single HCC of up to 3 cm in diameter, MWA was compared with resection. There were no statistically significant differences in the 1-, 3-, and 5-year OS rates (MWA: 96.2%, 80.3%, and 55.4%, respectively; resection: 91.3%, 81.4%, and 64.8%, respectively; HR = 1.06; 95% confidence interval [CI], 0.61–1.85; *p* = 0.839) and DFS rates (MWA: 72.4%, 43.2%, and 26.4%, respectively; resection: 78.8%, 51.2%, and 38.0%, respectively; HR = 1.27; 95% CI, 0.84–1.90; *p* = 0.247) between the MWA and resection groups. Patients in the MWA group once again had fewer complications (52.5% vs. 97.5%, *p* < 0.001) and a shorter postoperative hospital stay (3 days vs. 6 days, *p* < 0.001) than those in the resection group [32].

When it comes to multifocal HCC, an interesting retrospective study has been performed in China. In a group of 289 resectable patients with multifocal 3–5 cm HCC, the authors compared laparoscopic resection with MWA. The median OS was 97.4 months in the laparoscopic resection group and 75.2 months (95% CI 47.8–102.6) in the MWA group during a follow-up period of 39.0 months. The 1-year, 3-year, and 5-year OS rates in the two groups were 91.8%, 72.6%, and 60.7% and 96.5%, 72.8%, and 62.5%, respectively. The corresponding DFS rates were 75.9%, 57.2%, and 46.9% and 53.1%, 17.5%, and 6.2% (HR = 0.35, 95% CI 0.23–0.54, *p* < 0.001) [33]. The resection was therefore superior to MWA in terms of the recurrence rate but it did not influence OS. The difference in local recurrences was less pronounced in more recent cases, but it still showed the superiority of resection over MWA in the context of local recurrence, with no negative effect on OS.

Local recurrence of HCC after resection or after radical ablation remains an important clinical problem. Recurrence after resection of HCC can reach 60 to 80% after 5 years [34]. Furthermore, as stated above, the risk of local recurrence after RFA and MWA ablation techniques can be higher than 70% after 3 years [28]. The decision regarding the type of treatment after local failure of resection or ablation is still not well defined. Multifocal recurrence is an indication for systemic treatment as it may reduce the tumour burden and eradicate micrometastatic disease, which can, in turn, decrease recurrence rates [35]. Recurrence in patients that present with BCLC stage C is also an indication for systemic treatment [36]. Unresectable local recurrences are candidates for transarterial chemoembolisation [36].

The pattern of recurrence is also an important factor that influences post-recurrence survival. It has been shown that patients with local recurrence after RFA for HCC have a better prognosis than patients with non-local recurrences [37]. It is unclear, however, whether non-local recurrence is a result of previously undiagnosed subclinical metastatic foci or de novo metastatic disease. It seems that one of the crucial elements that could potentially minimise the risk of recurrence is systemic therapy [38].

Any type of curative intervention, be it ablation, liver resection, or salvage transplantation, can improve outcomes in patients experiencing recurrent HCC. The curative treatment of single local metastasis with re-resection or re-ablation—if at all possible—yields substantially better results than a non-curative approach. In a group of 544 patients with local recurrence after radical (R0) resection of an HCC, a curative approach including resection or ablation yielded better survival than a non-curative approach that included systemic chemotherapy and transarterial chemoembolisation [36].

The role of systemic therapy in preventing local and distant failure after curative therapies for HCC is unclear. However, several studies suggest that, particularly in patients with aggressive tumour biology, systemic therapies have significant potential [38].

## 3. Future Directions for Ablative Therapies in CLM and HCC

When discussing these highly promising results of ablative therapies in the context of CLM treatment, one has to remember that the bulk of data come from highly experienced HPB centres. The results of ablation seen in those centres are not necessarily easy to reproduce in smaller hospitals with less or even no experience with the surgical management of CLMs. Therefore, in order to maintain the high quality of ablation and resulting favourable oncological outcomes, specialised programs designed to introduce and popularise the appropriate use of this technique will be required [24].

The role of ablative therapies in the treatment of small HCC as a bridge therapy for LT is firmly established [10]. The European Association for the Study of the Liver (EASL) described that the MWA technique showed promising results in terms of local control and survival in HCC patients [39]. As shown in the data cited above, it seems that in the case of larger tumours, the impact that ablation has on survival is comparable with resection, but local control is still somewhat inferior [33]. As experience with MWA increases, which, at least theoretically, should be superior to RFA in its biological local tissue impact [1,16], and as higher-power ablation is employed [30], we can hopefully see even better results for larger and multifocal tumours [33]. The best treatment options for HCC remain LT and resection [10]. Unfortunately, organ shortage remains the main obstacle to the wider use of LT for HCC treatment. In a recent series, only 2.6% of BCLC stage A patients with a solitary HCC of 3 to 5 cm that were initially treated with RFA eventually received an LT [8]. However, with the highly encouraging safety profile of ablative therapy, improvement of its local control and long-term oncological results can make it a serious contender for the title of the best treatment for HCC. In the current version of the BCLC algorithm, ablation is offered to patients falling into very early (0) and early (A) stages of HCC who are not otherwise candidates for surgery. Based on the previously cited studies, it seems that ablation will continue to gain more ground over surgical resection, mainly due to its safety profile.

An important element of the further development of ablative therapies for CLMs and HCC is quality assurance. The equivalent of R0 surgical resection in the context of ablation is A0 ablation, which translates into adequate safety margins exceeding 5 mm on an appropriate imaging technique [40]. This element of ablative treatment is crucial in order to minimise local recurrence risk. The re-treatment of local recurrences, whenever possible, has a better prognosis than the treatment of non-local recurrences [37], but minimising the risk of recurrence is still the best available strategy, hence the importance of quality control in ablation. The correct use of proper imaging techniques (a combination of ultrasound, CT, and MRI) in preparative planning and in postoperative verification of A0 ablation margins is therefore crucial for quality assurance [1].

Arguably one of the most promising and most active areas of research in the further development of ablative therapies is the use of artificial intelligence. AI may prove to be crucial in calculating the optimal point of entry and angle of attack of the ablation needle, as it could play a pivotal role in assessing A0 radiological radicality [40]. A machine-enhanced learning tool for the analysis of MRI images has already proved to be superior to standard clinical features in predicting local recurrence after ablation of HCC [41]. If confirmed, the use of such AI tools could help to identify patients who could be ideal candidates for adjuvant therapy.

## 4. Conclusions

Ablation therapy for liver lesions in the form of RFA and MWA is a useful tool in the treatment of CLMs and HCCs. Its use mandates adequate preoperative planning, training, and expert use of imaging techniques. It is quickly gaining ground in the treatment of CLM, mainly due to its consistently favourable safety profile. In the treatment of HCC, it is well established as a bridge therapy to LT, but recent data suggest its favourable long-term oncological results, with a local recurrence profile that is only slightly inferior compared to resection. Ablation also seems to have significant potential in the treatment of local recurrences of HCC.

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
