# Peer review of "The Emerging Role of Ablation in the Treatment of Primary and Metastatic Cancer of the Liver"

_jcm, 2025, doi:10.3390/jcm14228016_

Round 1

Reviewer 1 Report

Comments and Suggestions for Authors

The review is really too short! Only 1 page on the ablation of HCC is not enough for a "comprehensive review"!!! I suggest to add several comments, for example a table summarizing the studies and/or meta-analyses comparing RFA vs MWA in HCC patients.

The autho should also add a paragraph on the tested adjuvant therapies after RFA/MWA, for example citing the relevant series PMID: 25974743)

The authors should comment also on the concept of post-recurrence survival in these patients

Some figures would improve the quality of the manuscript

Author Response

Comment 1: The review is really too short! Only 1 page on the ablation of HCC is not enough for a "comprehensive review"!!! I suggest to add several comments, for example a table summarizing the studies and/or meta-analyses comparing RFA vs MWA in HCC patients.

Response: Several paragraphs has been added (marked clearly in yellow)

Comment 2: The autho should also add a paragraph on the tested adjuvant therapies after RFA/MWA, for example citing the relevant series PMID: 25974743)

Response: A relevant paragraph has been added. 10 new references have been added as well to clearly explain all points made

Comment 3: The authors should comment also on the concept of post-recurrence survival in these patients

Response: A relevant paragrpah with references has been added

Comment 4: Some figures would improve the quality of the manuscript

Response: The author thinks all the points re clearly explained without the use of figures. If the Reviewer feels that even in its current version the manuscript requires a figure/table, it will be provided

Reviewer 2 Report

Comments and Suggestions for Authors

This review briefly discusses the overall place of ablation therapies in the current clinical practice schematic for treating hepatic malignancies (specifically colorectal mets and priamary hepatocellular carcinoma). There are a few point within the text that should be addressed prior to publication:

  • The introduction does not clearly identify the novelty, need, nor overview of the current review.
  • The comment on Lines 95-97 about the heat sink impact on MFA is based on a weak reference (a review article, that cites a review article, that doesn’t site anything). This should either have a more appropriate citation, or the example should not be used to show that MWA is different than RFA.
  • Lines 237-240, the comment on AI is within the HCC section, but it is not specific. This should be elaborated on, and either written to be HCC specific or placed into a separate AI in ablation section.
  • Line 40 “With growing number of evidence,” Line 79 “we see a very good oncological outcomes,” Line 161 “In a group of patients reaching almost 30% experiencing pathological complete response” are a few examples of awkward phrases throughout the manuscript that can use grammar improvements to improve readability

Author Response

Comment 1: The introduction does not clearly identify the novelty, need, nor overview of the current review.

Response: The introduction has been re-written. New elements are marked in yellow

Comment 2: The comment on Lines 95-97 about the heat sink impact on MFA is based on a weak reference (a review article, that cites a review article, that doesn’t site anything). This should either have a more appropriate citation, or the example should not be used to show that MWA is different than RFA.

Response: The cited reference number 16 in the current version is an experimental study comparing biological effects of MWA, RFA and a new experimental probe.

Comment 3: Lines 237-240, the comment on AI is within the HCC section, but it is not specific. This should be elaborated on, and either written to be HCC specific or placed into a separate AI in ablation section.

Response: The comment has been elaborated, one reference added and it is placed under the heading "3. What is the future for ablative therapies of CLM and HCC?"

Comment 4: Line 40 “With growing number of evidence,” Line 79 “we see a very good oncological outcomes,” Line 161 “In a group of patients reaching almost 30% experiencing pathological complete response” are a few examples of awkward phrases throughout the manuscript that can use grammar improvements to improve readability

Response: All the phrases pointed by the Reviewer were re-written

Round 2

Reviewer 1 Report

Comments and Suggestions for Authors

The revised paper is OK

Reviewer 2 Report

Comments and Suggestions for Authors

Thank you for making adjustments, the manuscript is improved from previous version. No further scientific suggestions at this time. 

Comments on the Quality of English Language

There are improvements from the previous version, but additional edits would be beneficial to improve clarity.